# Daily Living Activity Recognition with Frequency-Shift WiFi Backscatter Tags [note 1]

**DOI:** 10.3390/s24113277

**Published:** 2024-05-21

**Authors:** Hikoto Iseda, Keiichi Yasumoto, Akira Uchiyama, Teruo Higashino

**Affiliations:** 1Department of Science and Technology, Nara Institute of Science and Technology, 8916-5 Takayama-cho, Ikoma 630-0192, Japan; 2RIKEN Center for Advanced Intelligence Project AIP, 1-4-1 Nihon-bashi, Tokyo 103-0027, Japan; 3Graduate School of Information Science and Technology, Osaka University, 1-5 Yamada-oka, Suita 565-0871, Japan; uchiyama@ist.osaka-u.ac.jp; 4Faculty of Engineering, Kyoto Tachibana University, 34 Yamada-cho, Oyake, Yamashina-ku, Kyoto 607-8175, Japan; higashino-t@tachibana-u.ac.jp

**Keywords:** frequency-shift backscatter tag, WiFi, living activity recognition

## Abstract

To provide diverse in-home services like elderly care, versatile activity recognition technology is essential. Radio-based methods, including WiFi CSI, RFID, and backscatter communication, are preferred due to their minimal privacy intrusion, reduced physical burden, and low maintenance costs. However, these methods face challenges, including environmental dependence, proximity limitations between the device and the user, and untested accuracy amidst various radio obstacles such as furniture, appliances, walls, and other radio waves. In this paper, we propose a frequency-shift backscatter tag-based in-home activity recognition method and test its feasibility in a near-real residential setting. Consisting of simple components such as antennas and switches, these tags facilitate ultra-low power consumption and demonstrate robustness against environmental noise because a context corresponding to a tag can be obtained by only observing frequency shifts. We implemented a sensing system consisting of SD-WiFi, a software-defined WiFi AP, and physical switches on backscatter tags tailored for detecting the movements of daily objects. Our experiments demonstrate that frequency shifts by tags can be detected within a 2 m range with 72% accuracy under the line of sight (LoS) conditions and achieve a 96.0% accuracy (F-score) in recognizing seven typical daily living activities with an appropriate receiver/transmitter layout. Furthermore, in an additional experiment, we confirmed that increasing the number of overlaying packets enables frequency shift-detection even without LoS at distances of 3–5 m.

## 1. Introduction

To provide context-aware services such as context-aware home appliance control, elderly care monitoring systems, and health support applications, it is essential to recognize the activities of daily living (ADLs) of residents in homes. Methods for recognizing daily activities can be broadly categorized into two approaches: those that require wearing devices such as smartphones and wearable devices [1,2,3,4,5,6,7,8,9], and those that involve installing non-contact sensors in the environment [10,11,12,13]. The former necessitates the constant wearing of sensing devices. This method is generally capable of tracking fine-grained bodily movements with high accuracy, and effective feature extraction techniques have been established, as exemplified by Hartmann et al. However, it poses a significant physical burden on users [14]. The latter, while avoiding the need for worn devices, incurs high costs for installation and maintenance and raises privacy concerns when cameras or microphones are employed. To overcome these challenges, recent proposals have explored the use of energy-harvesting devices [15,16,17,18]; however, these methods share similar drawbacks with traditional approaches and highly depend on the availability of energy-harvesting conditions.

In light of these issues, radio wave-based activity recognition technologies, such as WiFi Channel State Information (CSI), RFID, and backscatter communication, have garnered attention. The approach using WiFi CSI exploits the variations in channel state information caused by human activities through reflection, refraction, and scattering [19,20,21,22,23]. This method only requires a transmitter and receiver, facilitating non-contact activity recognition with low installation and maintenance costs. However, it faces a fundamental limitation: models trained in specific environments are not applicable in others. The use of passive RFID for activity recognition, which leverages the fluctuations in radio waves caused by human activities and interaction with RFID tags [24,25,26,27], also faces challenges related to environmental dependency and the need for specialized equipment like RFID readers capable of detecting multiple RFID tags simultaneously, a feature not commonly found in general households, making practical implementation in real-life settings difficult. Backscatter communication, which encodes data in reflected waves by altering the reflection/absorption state in response to ambient radio waves [28], represents a promising direction. Existing backscatter communication methods for activity recognition [29,30,31] typically employ a system comprising a transmitter (Tx), backscatter tags for communication, and a receiver (Rx). These systems primarily detect activities through changes in radio waves caused by human actions or convert physical movements into communicable actions via physical switches. Backscatter communication has the advantage of low installation and maintenance costs by utilizing ambient radio waves and existing communication infrastructures like WiFi. However, most existing methods require continuous wave emission, leading to potential interference with other communication channels and imposing significant limitations on the detection ranges of tags. To the best of our knowledge, no study has discussed the feasibility of in-home activity recognition through backscatter communication in environments crowded with numerous wireless devices, walls, home appliances, and other radio wave obstructions.

In this paper, we aim to investigate the feasibility of backscatter communication for recognizing ADLs in homes, leveraging its benefits of low installation and maintenance costs and low privacy exposure while addressing the limitations of existing methods. We employ the frequency-shift WiFi backscatter tag developed by Nakagawa et al. [32], referred to herein as the “backscatter tag” or simply the “tag”. Unlike existing methods that often involve transmitting sensor data or functioning as relay devices, the frequency-shift backscatter tag is designed solely for sensing using the tag. It consists of a compact battery, an oscillator, an RF switch, and a physical switch. When the physical switch is activated, the oscillator generates a specific frequency, causing the RF switch to toggle on and off, thereby inducing a frequency shift in the reflected waves. The oscillator can be set to any frequency within the 0–10 MHz range, allowing for the detection of the corresponding physical switch’s on/off status by examining the observed frequency shift’s bandwidth. We utilize SD-WiFi, a software-defined WiFi access point, as the communication foundation, which can transmit packets containing Continuous Wave (CW) signals as part of the payload. While such specifications are indeed not found in conventional WiFi devices, unlike RFID readers which are not commonly deployed in most households, WiFi devices for transmitting and receiving are already widespread in general households. This makes them well-suited as a foundation for recognizing daily activities in residential environments. Also, unlike traditional methods, SD-WiFi enables discrete communication on a per-packet basis, facilitating coexistence with various other signals in real residential environments. Moreover, even if communication disruptions result in packet loss, intermittent communication allows for tag detection, addressing challenges associated with conventional backscatter communication-based activity recognition methods.

The aim of this study is to discuss the potential of frequency-shift backscatter tags for recognizing daily living activities in typical home environments. Specifically, by installing SD-WiFi and backscatter tags in actual residential settings, we aim to explore the answers to the following three research questions (RQs):**RQ1**:When SD-WiFi is installed in a typical living environment, how accurately can the on–off states of backscatter tags be detected?**RQ2**:When tags are attached to daily living objects involved in various ADLs, how accurately can on–off states of tags be detected?**RQ3**:How accurately can various ADLs be recognized by detecting on–off states of tags attached to daily objects?

To answer RQ1, we first implement a system using frequency-shift backscatter tags in our smart home testbed, constructed to reproduce a typical living environment. We investigate the detection accuracy of the tags’ on–off states by changing the distance between the SD-WiFi transmitter (Tx) and receiver (Rx) and the positions of backscatter tags.

To answer RQ2, we attach tags to daily living objects involved in ADLs (doors, faucets, chairs, bedding, etc.) and investigate the feasible placement of SD-WiFi transmitters and receivers that allows efficient acquisition of the tags’ on/off states. Additionally, we adjust the tags’ switches to facilitate easier on–off switching during interaction between the residents and the objects.

To answer RQ3, we conducted three types of ADL recognition experiments: Experiment 1, which involved three participants performing activities in a fixed sequence and duration; Experiment 2, an in-the-wild experiment with one participant, where both the sequence and duration of activities were variable; and Experiment 3, where the sequence and duration of activities were variable in a virtually altered space. In Experiment 1, using leave-one-session-out cross-validation, we achieved an average F-score of 0.933, and with leave-one-person-out cross-validation, the average accuracy was 0.926. For the in-the-wild activity recognition experiment (Experiment 3), where neither the sequence nor the duration of activities were predefined, the leave-one-session-out cross-validation resulted in a maximum accuracy of 0.955. In Experiment 2, where the locations of the toilet and bathroom were virtually swapped, the leave-one-session-out cross-validation achieved an accuracy of 0.971. These results demonstrate the effectiveness of our method.

Our previous work [33] proposed the basic idea of the WiFi backscatter-based ADL recognition system and showed preliminary experiment results. This paper thoroughly extends the previous work in the following points:We refined the ADL recognition algorithm and provided a more detailed analysis.We designed and conducted additional experiments reproducing more realistic daily living environments.We designed and developed a new method that overlays more packets to extend the distance between backscatter tags and WiFi transceivers capable of detecting tags’ on–off states and showed that the distance could be extended to up to 5 m compared to the previous work (1–2 m).

## 2. Related Work

This section reviews existing methods for in-home activity recognition using radio waves, categorized into approaches that utilize WiFi, Radio-Frequency Identification (RFID), and backscatter communication techniques.

### 2.1. WiFi-Based Methods

WiFi-based methods are device-free and leverage existing communication infrastructure already installed in many homes. Most methods using WiFi utilize Channel State Information (CSI), which provides amplitude and phase information of WiFi signals on each subcarrier. Human activities cause variations in WiFi signals through reflection, refraction, and scattering, which are reflected in the CSI, enabling activity recognition. Yan et al. collected CSI data using a commercial single WiFi access point, profiling different characteristics for seven activities including eating, bathing, and handwashing, achieving an average recognition accuracy of 0.92 [19]. Li et al. also used both amplitude and phase information extracted from CSI sequences to distinguish activities like bending and jumping, achieving an accuracy of 0.966 [34]. Moshiri et al. set up a WiFi CSI data collection environment using Raspberry Pi and applied CNN to data represented as 2D images, recognizing activities like walking, running, and standing up with an accuracy of about 0.95 [20]. Shang et al. used a unique machine learning architecture combining CNN and LSTM to distinguish activities like walking, running, and standing up with an average accuracy of 0.95 [21]. Shang et al. used a unique machine learning architecture combining CNN and LSTM to distinguish activities like walking, running, and standing up with an average accuracy of 0.95 [21]. Jannat et al. present a Wi-Fi-based human activity recognition system that uses an Adaptive Antenna Elimination algorithm, achieving accuracies up to 99.84% on activities like walking, falling, and sitting across different environments [22]. Ding et al. introduced a device-free human activity recognition system utilizing a deep complex network that processes both amplitude and phase of Wi-Fi signals. Tested in an office environment, this method demonstrated high accuracies of 96.85% and 94.02% across 24 locations for five distinct activities [23].

### 2.2. RFID-Based Methods

Passive RFID, which does not require power for communication, offers maintenance-free activity recognition similar to WiFi. Jin et al. [24] developed a method to track movements by attaching passive RFID tags to various parts of clothing and using a single RFID reader, achieving an absolute error of 8–12 degrees in estimating the posture of single-degree-of-freedom joints like elbows and knees, and a bearing error of 21° and an elevation error of 8° for two-degree-of-freedom joints like shoulders. Wang et al. [25] used passive RFID tags embedded in clothing and an RFID reader attached to the body to recognize eight different activities in real-time, including sitting, standing up, and walking, achieving an accuracy of 0.963. Moreover, Wang et al. [27] placed RFID tags in the environment and successfully recognized eight types of activities, including walking, sitting, and turning, with an accuracy of 0.935 by utilizing the variations in reflected signals caused by human activity.

### 2.3. Backscatter-Based Methods

Backscatter communication, operating on ultra-low power by utilizing ambient radio waves and eliminating the need for the device itself to emit radio waves, has garnered significant attention for maintenance-free activity recognition. Zhang et al. [35] designed a system comprising frequency-shift backscatter tags and transceivers worn on the body, demonstrating the potential of wearable devices using backscatter tags for human activity recognition. Li et al. have developed an activity recognition system named Back-Guard [29] comprising small Tx, Rx, and passive backscatter tags. Back-Guard generates frequency-shifted signals with backscatter tags in response to continuous waves transmitted from Tx using the changing characteristics of these signals due to human actions, achieving an activity recognition accuracy of 0.934 and a user identification accuracy of 0.915 for 25 subjects. BARNET [30] proposed by Ryoo et al. realizes backscatter communication between tags and utilizes the nature of changing Backscatter Channel State Information (BCSI) due to human actions for activity recognition. BARNET can recognize approximately eight basic daily activities, such as brushing teeth, running, walking, and sitting, with an error rate of about 0.06. Printed WiFi [31] uses 3D-printed conductive materials to create antennas and gears, enabling backscatter communication over WiFi by associating the on/off state of an RF switch with physical movements like wind, water flow, or button operations. Printed WiFi can detect signals from up to 17 m away.

### 2.4. Challenges of Existing Methods and Position of This Study

Methods using WiFi or RFID waves can utilize infrastructure already installed in homes, offering the advantage of device-free activity recognition. However, these WiFi-based methods are heavily influenced by the environment, necessitating learning for each specific setting. RFID-based methods offer a wide detection range and maintenance-free operation but necessitate specialized readers capable of detecting multiple RFID tags simultaneously, a feature not commonly found in general households, thus limiting their practical deployment in residential settings.

Furthermore, methods using backscatter technology, while requiring tags for communication, do not need additional power sources and can capture fine human movements. However, most existing methods rely on continuous waves, making coexistence with other communication channels difficult, and have strong limitations on the detectable movement locations. Moreover, The feasibility of backscatter communication-based activity recognition methods in noisy environments, such as general households with various radio waves, walls, furniture, appliances, and other obstacles that hinder communication, has not been sufficiently discussed. Therefore, this study aims to investigate the feasibility of backscatter communication for activity recognition in noisy environments, considering real-life living conditions.

## 3. Daily Living Activity Recognition System Using Frequency-Shift WiFi Backscatter Tags

This section details the proposed system for recognizing ADLs using frequency-shift WiFi backscatter tags. The overall architecture of the proposed system is depicted in Figure 1, consisting of (1) a group of **backscatter tags** used to detect movements of furniture, appliances, doors, and drawers; (2) **SD-WiFi AP groups** for transmitting and receiving WiFi radio signals; (3) **SD-WiFi Cloud** for storing signal data received by APs and collecting packets communicated between SD-WiFi, and; (4) an **activity recognition module** that detects frequency shifts, extracts features, and performs activity recognition.

### 3.1. Frequency-Shift Backscatter Tags

The frequency-shift backscatter tag, as shown in Figure 2, consists solely of an antenna, oscillator, physical switch, and battery. The physical switch toggles the circuit on/off in response to movements of furniture or appliances. When power is supplied to the circuit, the oscillator generates a signal corresponding to its preset variable resistor. The oscillator is directly connected to the RF switch, and the on/off toggling of the RF switch in response to vibrations changes the impedance of the antenna, thereby inducing a frequency shift in the scattered waves.

In an environment where some carrier wave exists, when the backscatter tag conducts (the physical switch is on), it rapidly toggles the RF switch on/off At the moment the RF switch is on, the wave captured by the backscatter tag is reflected as is, meaning the amplitude of the carrier wave is transmitted to the receiver unchanged. However, when off, it is absorbed and not reflected, making the amplitude of the reflected wave zero. Thus, the backscatter tag modulates the amplitude of the external carrier wave via the antenna [32].

Assuming the frequency of the carrier wave present in the environment is *f*, and the frequency set on the oscillator of the *i*-th backscatter tag is $f_{i}$, such amplitude modulation can be expressed as the product of two waves and is hence decomposable into two cosine waves with frequencies $f-f_{i}$ and $f+f_{i}$:(1)$$2\left(\sin2\pi ft\right)\sin\left(2\pi f_{i}t\right)=\cos\left(2\pi \left(f-f_{i}\right)t\right)-\cos\left(2\pi \left(f+f_{i}\right)t\right)$$

The above equation shows that the presence of a backscatter tag in the environment causes frequency shifts of $f-f_{i}$ and $f+f_{i}$ relative to the carrier wave *f*. Observing the presence or absence of these frequency shifts allows us to determine the on/off status of the physical switch of tag *i*. Therefore, pre-registering the tag and its shift width in a database makes it possible to detect movements of tags attached to furniture, appliances, and doors without complex computations.

The frequency-shift backscatter tags used in this study are designed to be extremely simple, thereby consuming very low power. According to Nakagawa et al., the total power consumption of the primary components of the backscatter tag, the RF switch, and oscillator, is 11 μA, with power consumption on the order of μW. Therefore, the backscatter tag is expected to operate for approximately two years without sleep mode when powered by a CR2032 coin cell (225 mAh), and semi-permanent operation is anticipated with the application of energy harvesting technology [32].

### 3.2. SD-WiFi AP/Cloud

In this system, a software-defined radio base station, the SD-WiFi AP, is used to generate WiFi signals. The appearance of the SD-WiFi is illustrated in Figure 3. Through software control, the SD-WiFi is capable of communicating on any frequency within the WiFi frequency band. Communication occurs on a packet-by-packet basis, with control information contained within the packet header and a sine wave within the payload. This specification is due to the fact that frequency shifts in backscatter tags occur in response to all surrounding radio waves, while the signals of conventional WiFi contain various frequency components simultaneously, making it difficult to observe frequency shifts. By embedding a single frequency sine wave in the payload, it becomes easier to detect frequency shifts. These frequency shifts appear before or after this sine wave, and observing either is sufficient, allowing for robust operation even in noisy environments with numerous radio waves.

The SD-WiFi Cloud acquires packets from the SD-WiFi AP and is connected through a router on the same network as the SD-WiFi AP. The SD-WiFi can acquire packets from the SD-WiFi AP at intervals as short as 0.25 s.

### 3.3. Activity Recognition Module

First, the information of packets affected by frequency shifts in the space where backscatter tags are present, namely the raw sine wave signals, are extracted from the SD-WiFi AP/Cloud. The raw sine wave signals can be obtained by dividing them into *I* and *Q* waves as shown in Figure 4, and these are combined to extract a single sine wave. Specifically, with the *I* wave represented as I(t), the *Q* wave is Q(t),  the frequency is *f*, and the raw signal s(t) is calculated according to the following equation:(2)s(t)=I(t)cos(2πft)−Q(t)sin(2πft)

Then, Fast Fourier Transform (FFT) is performed on the raw signals to confirm the frequency shifts. Backscatter communication is a technique that carries data on reflected waves, but these reflections are subject to various noises from the environment, resulting in multiple peaks in the FFT results. Therefore, it is difficult to distinguish between frequency shifts and noise from the FFT results of a single packet. However, unlike noise, the reflections can be continuously received while backscatter communication is occurring. This characteristic is leveraged by overlaying the FFT results of multiple packets to observe frequency shifts.

Now, let $V_{i}$ be the vector storing the results of the Fast Fourier Transform (FFT) applied to the *i*th packet. Since all packets without errors have vectors of the same length, the vector W resulting from overlaying *k* packets is calculated using the following equation:(3)$$W=\sum_{i=1}^{k}V_{i}$$

A visualization of the results of the process is shown in Figure 5, where the frequency of the sine wave is set to 5.0 MHz, the frequency-shift width on the tag is set to 1.0 MHz, the backscatter tag between APs is placed to generate a frequency shift, and the results of 10 packets are overlaid. Based on Figure 5, it is evident that there is a peak at the position indicated by the red arrow, exactly 1.0 MHz away from the frequency of the sine wave.

To observe this frequency shift, the peak detection algorithm shown in Algorithm 1 is employed. Prior to applying the peak detection algorithm, leveraging the characteristic that allows presetting of the frequency-shift width for each tag, a bandpass filter is applied to isolate the predetermined frequency range. array[] represents the results of the FFT (only near the target frequency), and treshold is a coefficient representing the difference between peaks and non-peaks. In this peak detection algorithm, the moving average of *N* elements of the array, average, is calculated and if the moving average of the next element is greater than the value of average multiplied by threshold, it is considered that a peak exists and returns True. If a peak is not found, it returns False.
**Algorithm 1** Peak detection algorithm  1:**function** Peak_detection(array,N,threshold)  2:    flag←False  3:    **for** i=0 to array.length−N−1 **do**  4:        $average\leftarrow \frac{1}{N}\sum_{k=i}^{i+N}array\left[k\right]$  5:        **if** a[i+1]>average×threshold **then**  6:            flag←True  7:            **break**  8:        **end if**  9:    **end for**10:    **return** flag11:**end function**

### 3.4. Algorithm for Activity Recognition

Each backscatter tag can be set with a different frequency-shift width. Here, the frequencies set for each tag are denoted as $f_{i}\;\left(i=1,\ldots ,n\right)$, where *n* is the number of different frequency-shift widths. At any given moment in the received signal, if a peak is observed in the specific frequency space corresponding to $f_{j}$, it indicates that the furniture or appliance associated with that tag has been used. By encoding the presence or absence of frequency shifts as binary data (1/0), it is possible to acquire time series data of tag movement.

Time series data are extracted using a window function, defining a window size *w*. These data are flattened into one dimension and input into a machine learning model, which is then trained to associate the movements of the tags with corresponding actions. For example, suppose tags are placed such that they turn on during actions like opening the refrigerator or lifting the faucet handle, while actions in the kitchen such as “dish washing” or “cooking” may be identified, if the refrigerator tag frequently triggers, it likely indicates “cooking”, and if the faucet tag triggers without the use of the refrigerator for an extended period, it likely indicates “dish washing”. The model learns to associate such furniture and appliances with specific actions.

## 4. Preliminary Experiments

This section aims to answer RQ1 and RQ2 by conducting two experiments in a typical residential environment. Specifically, we perform the following two experiments:**Preliminary Experiment 1**: An experiment to confirm the accuracy of frequency shift-detection when varying the distance between the SD-WiFi and backscatter tags.**Preliminary Experiment 2**: An experiment to confirm the accuracy of frequency shift-detection when backscatter tags are attached to household objects such as furniture, appliances, and doors, assuming in-home action recognition.

These experiments were conducted at the NAIST Smart Home. The interior of the NAIST Smart Home is as shown in Figure 6.

### 4.1. Preliminary Experiment 1

This experiment aims to investigate whether the on/off states of frequency shifts can be detected in a real residential environment. For the experiment, a pair of SD-WiFi APs (Tx and Rx) were placed on the floor of the smart home, and the experiment was conducted at distances of 1 m and 2 m, respectively. The tag is equipped with a tilt switch, activating a frequency shift when tilted. As shown in Figure 7, the tags were positioned in a straight line between Tx and Rx, spaced 25 cm apart, with antennas aligned towards Tx. During the experiment, packets were transmitted every second for 60 s at each tag’s position when it was standing vertically (on) and lying horizontally (off), and this process was repeated in three sessions to collect data.

The collected data were analyzed using the peak detection algorithm to evaluate the detection performance of frequency shifts. The number of packets overlaid is set to 10 packets.

Table 1 and Table 2 show the F-scores for the detection rate of the tags. From these tables, it is evident that at a distance of 1m between the APs, frequency shifts are detected with high accuracy regardless of the distance of the tags. However, when the distance between the APs is 2 m, it is observed that the detection rate is relatively high (0.776–1.0) when tags are placed close to Tx (0.25–0.75 m) or Rx (1.75 m) but decreases (0.416–0.62) as the distance from Tx or Rx increases (1.00–1.50 m). A possible reason for the reduced accuracy at the midpoint, where the distance between the tag and APs is the greatest, could be due to effects such as fading.

### 4.2. Preliminary Experiment 2

The aim of this experiment is to evaluate the detection accuracy of the on/off states of frequency-shift backscatter tags attached to household objects relevant for in-home activity recognition. Results from Experiment 1 suggest that for backscatter tags to be detected by the SD-WiFi, they need to be positioned relatively close to either Tx or Rx, and a line of sight (LoS) between the tag and AP is necessary. Therefore, pairs of SD-WiFi APs were installed in each room of the smart home, ensuring these two conditions were met. The specific arrangements of the SD-WiFi and backscatter tags are shown in Figure 8.

Backscatter tags were installed as follows: In the hallway, tags were placed on the doors of the bathroom and changing room; in the kitchen, tags were placed on the faucet and the refrigerator door; in the living room, tags were placed on the remote control and the back of a chair; and in the bedroom, tags were placed on the bedroom door and the bed. The arrangement of the backscatter tags is shown in Figure 9. Each tag is equipped with a special physical switch, designed to turn on during events such as when a door is opened, someone sits on a chair, or a faucet is turned on. For each tag’s location, the height of the SD-WiFi AP was varied using adjustable shelves from the floor, and packets were continuously transmitted for 100 s, with data collected over three sessions.

The results are presented in Table 3. Due to physical factors such as shelves and walls, measurements for some tags were not conducted. As a general trend shown in Table 3, it is evident that detection rates are better at heights above 0.75 m. This can be attributed to the reduction in radio wave reflections from surrounding walls and floors by changing the height, making it easier to observe frequency shifts.

## 5. Evaluation Experiment

In this section, to answer RQ3, we conduct actual ADL recognition experiments and construct a living activity recognition model in the NAIST Smart Home, which replicates a real living environment, and evaluate its accuracy. Specifically, we perform the following three experiments:**Experiment 1**: An ADL recognition experiment with activities and timings somewhat fixed, based on action scenarios by three participants.**Experiment 2**: An in-the-wild ADL recognition experiment with one participant, where the order and timing of activities are variable.**Experiment 3**: An ADL recognition experiment with variable order and timing of activities, including virtual swapping of spaces.

While Experiment 1 is the most primitive, Experiments 2 and 3 aim to replicate more realistic living behavior patterns.

### 5.1. Data Collection and Experimental Environment

The experiments were conducted in the smart home with up to three participants. Backscatter tags and SD-WiFi APs were placed in the locations shown in Figure 8, similar to Preliminary Experiment 2. The targeted daily activities and their definitions are presented in Table 4. The selection of daily activities was based on the primary activities listed in the Ministry of Health, Labour and Welfare’s Survey on Time Use and Leisure Activities [36], aiming to cover as many activities as possible.

Data related to these seven activities were collected based on controlled scenarios. The scenarios were conceptualized as a series of activities from returning home to going to sleep, consisting of cooking, eating, dishwashing, watching TV, toileting, bathing, and sleeping in that order. The locations of the SD-WiFi APs were set based on the optimal positions derived from Preliminary Experiment 2.

The annotation of these activities was conducted in real-time using a custom-built smartphone application. Due to the constraints on AP installation locations and backscatter tag placement, data collection was divided into three scenarios for the following activities. The activity data for each scenario were collected over three sessions, and during analysis, data from each session were concatenated to replicate a series of activities from returning home to going to sleep.

#### 5.1.1. Data Collection Method for Experiment 1

The goal of Experiment 1 was to confirm if activity recognition is feasible in the most primitive setting. Specifically, data were collected following the scenarios shown in Table 5. This scenario is designed to simulate the sequence of activities from returning home to going to sleep. There were three participants (three males; 24±1 years, 176±2.65 cm) and each was instructed to perform each activity for approximately 5 min in accordance with the scenario. Data were collected over three sessions, with the scenario remaining the same across all sessions.

#### 5.1.2. Data Collection Method for Experiment 2

The objective of Experiment 2 is to confirm if ADL recognition is possible in more realistic situations where the order of activities changes or the duration of activities varies. The experiment was conducted over three sessions, and in each session, data were collected according to the order and duration of activities as shown in Table 6. Data collection was performed for each scenario within each session, and the data were concatenated to replicate a series of ADLs.

#### 5.1.3. Data Collection Method for Experiment 3

Experiment 3 aims to confirm whether the model can function in homes with different layouts. However, physically changing the layout of the smart home is challenging, so, the locations of the dressing room/bathroom and the toilet are virtually swapped. That is, activities typically performed in the dressing room/bathroom are conducted in the toilet space, and vice versa. The action scenarios are the same as in Experiment 2, and for parts of the activities that are unrelated to the room’s layout, the same data from Experiment 2 are used.

### 5.2. Physical Switches

It is crucial to determine how to generate frequency shifts to recognize a wide range of activities within the home, especially static activities. For example, using a door switch to observe when a door is open as someone passes through is desirable. To detect someone sitting on a chair, observing a frequency shift while the person is seated is necessary.

We devised and created three types of physical switches to meet these requirements. The appearance of the switches is shown in Figure 10. The combination of backscatter tag ID, the furniture it is installed on, and the physical switch attached is presented in Table 7.

#### 5.2.1. Push Button Switch

The push button switch is designed to generate a frequency shift while the button is being pressed. It utilizes a momentary button switch for this purpose.

#### 5.2.2. Tilt Switch

The tilt switch is designed to generate a frequency shift while the tag is tilted.

#### 5.2.3. Reed Switch

The reed switch is designed to generate a frequency shift only when the switch body is removed from the magnetic field of a magnet. A contactless-type reed switch is used. For example, when a door equipped with this switch is closed, the magnet attached to the door body comes into contact with the reed switch, causing the contacts to open and the circuit to turn off. When the magnet is removed, the contacts close, the circuit turns on, and a frequency shift is generated.

### 5.3. Preprocessing and Feature Extraction

Preprocessing and feature extraction were carried out in the following three stages. First, Fast Fourier Transform (FFT) was applied to the data obtained for each packet. The results of the FFT corresponding to each packet were aligned using timestamps, and the data were concatenated for each session. As four packets are transmitted per second and the SD-WiFi AP has two antennas, each timestamp contains up to eight data points. The FFT results of these eight packets were overlaid every second for real-time activity recognition, and frequency shifts were detected using the peak detection algorithm. In the vector representing the on/off states of all installed tags, elements corresponding to tags where peaks were detected are set to 1, and those where no peaks were detected are set to 0. Through these preprocessing steps, a matrix is generated with tags on the horizontal axis and time series on the vertical axis.

The preprocessed data is input into a machine learning model. First, the time window function divides time series data into segments of fixed time intervals. The dimension of the input data for the machine learning model is C×w, where *C* is the number of tags, and *w* is the window size (seconds). The matrix of size C×w is then flattened into a one-dimensional vector and is input into the machine learning algorithm.

For activity recognition, the random forest algorithm is used as the machine learning algorithm. The accuracy of activity recognition is evaluated by precision, recall, F1-score, and accuracy.

#### Introduction of Lag Features

It is known that daily living activities have certain patterns [37]. Therefore, considering not only the movement of furniture and appliances at each time but also their sequence could enable high-accuracy activity recognition. Thus, not only expressing the presence or absence of furniture and appliance movements in binary along the time series but also creating features that attenuate over time (Lag Features).

The lag feature is calculated as follows. First, $v_{t,i}$ represents the state at time *t* for the *i*-th tag, calculated by the recursive formula described below. $n_{t,i}$ indicates the state of whether a frequency shift was observed at time *t* for the *i*-th tag. Therefore, its value is binary, with 1 if a frequency shift is observed and 0 if not. With α(0<α<1) as the attenuation rate, $v_{t+1,i}$ is calculated by the following equation:(4)$$v_{t+1,i}=\min\left(1,\;\alpha \cdot v_{t,i}+n_{t+1,i}\right)\;\mathrm{where}\;\forall i,\;v_{0,i}=0$$

In Figure 11, we present the scenarios without the introduction of lag features (α=0.00) and with the introduction of lag features (α=0.99). This figure illustrates that the introduction of lag features results in more complex input features.

## 6. Experimental Results

This section presents the results of each experiment. In Experiment 1, which involved three participants, we performed both leave-one-session-out cross-validation using two sessions for training data and one session for validation data for each participant and leave-one-person-out cross-validation using two participants for training data and one participant for validation data. For Experiments 2 and 3, which involved a single participant, only leave-one-session-out cross-validation was conducted.

### 6.1. Experiment 1

#### 6.1.1. Leave-One-Session-Out Cross-Validation

First, the transition of the attenuation rate α and F-score when conducting leave-one-session-out cross-validation for each participant is shown in Figure 12. From Figure 12, it can be seen that the accuracy improves as the value of α or the window size increases. This is likely because larger values of α or window size allow for longer learning of past activity patterns. Additionally, the accuracy changes are very similar across all three participants, indicating that these trends do not depend on the participant. Furthermore, while results within the range of 0–0.99 are presented in Figure 12, it was observed that the accuracy significantly deteriorates when the value of α exceeds 0.99999999 or higher, dropping to a maximum of 0.81 when α is set to 1.0.

Additionally, with the window size fixed at 9 and α set to 0.0, 0.5, and 0.99, respectively, the results of the leave-one-session-out cross-validation are presented in Table 8. From Table 8, it can be seen that activity recognition is possible with an accuracy of approximately 0.594–0.63 when α is 0.0, approximately 0.656–0.687 when α is 0.5, and approximately 0.968–0.972 when α is 0.99.

Furthermore, Figure 13 shows the confusion matrices for each attenuation rate. From Figure 13, it can be observed that when α is 0.0 or 0.5, activities without much movement other than door actions, such as “Bathing”, “Toileting”, and “TV Watching”, tend to be confused with each other, while activities with continuous responses during the action, such as “Eating”, “Dishwashing”, and “Sleeping”, can be recognized with high accuracy. When α is 0.99, most activities can be recognized with high accuracy, but there is a tendency to slightly confuse “Cooking” with “Other”, which involves little to no tag movement, such as moving.

#### 6.1.2. Leave-One-Person-Out Cross-Validation

The results of the leave-one-person-out cross-validation with the attenuation rates α set to 0.0, 0.5, and 0.99 are shown in Table 9. From Table 9, it can be seen that activity recognition is possible with an accuracy (F1-score) of 0.524 when α is 0.0, 0.621 when α is 0.5, and 0.955 when α is 0.99. This shows a decrease in accuracy compared to the average F1-scores of all participants during the leave-one-session-out cross-validation, which are, respectively, 0.561, 0.648, and 0.972, indicating a decrease in accuracy.

Moreover, from Figure 14, which shows the accuracy (F1-score) of activity recognition for each attenuation rate, it can be confirmed that, similar to the leave-one-session-out cross-validation, accuracy significantly improves as α increases.

Furthermore, the confusion matrices for α set to 0.0, 0.5, and 0.99 are shown in Figure 15. From Figure 15, it is clear that similar trends to those of the leave-one-session-out cross-validation can be observed, indicating that there is no significant difference in the model’s tendencies among participants, meaning the model is not dependent on the subject.

### 6.2. Experiment 2

Leave-one-session-out cross-validation was performed for each attenuation rate α, with the window size fixed at 9, and the average of each evaluation metric is presented in Table 10. From Table 10, similar to the results of Experiment 1, it is observed that accuracy improves as the value of α increases. However, at α=0.0, the F1-score is approximately 0.56, roughly the same as in Experiment 1, but even when α is increased to 0.99, the accuracy is 0.961, which is lower than the maximum accuracy of Experiment 1. This suggests that the variability in action scenarios between sessions makes it more challenging to learn from past activity histories.

Figure 16 shows the accuracy of activity recognition for each α value when the window sizes are 1 s, 5 s, and 9 s. Similar to the results of Experiment 1, from Figure 16, it is evident that accuracy dramatically improves when the value of α exceeds 0.8, and accuracy increases with larger window sizes. This suggests that parameter settings that allow for a broad learning of past activity histories are advantageous for activity recognition, even in scenarios where the action sequence changes between sessions.

### 6.3. Experiment 3

Leave-one-session-out cross-validation was performed for each attenuation rate α, with the window size fixed at 9, and the average of each evaluation metric is presented in Table 11. From Table 10, similar to the results of Experiments 1 and 2, it is observed that accuracy improves as the value of α increases. However, at α=0.0, the F1-score is slightly lower at 0.54 compared to Experiments 1 and 2, but even when α is increased to 0.99, the accuracy reaches 0.971, which is nearly the same as in Experiments 1 and 2.

Figure 17 shows the accuracy of activity recognition for each α value when the window sizes are 1 s, 5 s, and 9 s. Similar to the results of Experiment 1, from Figure 17, it is evident that accuracy dramatically improves when the value of α exceeds 0.8, and accuracy increases with larger window sizes, yielding results consistent with Experiments 1 and 2. These findings indicate that the proposed method tends not depend on environmental conditions such as the layout of rooms.

## 7. Additional Experiments

Through preliminary and evaluation experiments, it has been demonstrated that backscatter tags can be detected in typical living environments and that activity recognition can be performed with high accuracy. However, the conditions of the preliminary experiments currently impose constraints on the placement of SD-WiFi, specifically, the distance between SD-WiFi units must be no more than 2 m, and the distance between the tag and the access point must be within 1 m.

Because frequency shifts cannot be separated from noise with a single packet, it is necessary to overlay multiple packets. On the other hand, it is conceivable that increasing the number of overlays makes it possible to detect even weaker frequency shifts. Therefore, this section investigates how many packets are required to overlay to detect frequency shifts when the SD-WiFi is placed 3 m and 5 m apart, respectively.

### 7.1. Experimental Environment

The experiment was conducted in the NAIST Smart Home. The SD-WiFi was placed in the positional relationship shown in Figure 18, with distances set at 3 m and 5 m apart, and additionally, for both opened and closed states of a wooden sliding door situated between them. The backscatter tags were placed at distances that bisect the distances between the SD-WiFi APs in ratios of 1:2, 1:1, and 2:1. A tilt switch was attached to the tags, and approximately 1500 packets were collected for each state where the tag was upright (frequency shift occurs) and laid down (frequency shift does not occur). As shown in Figure 19, the experimental setup utilized a movable shelf to position both the SD-WiFi and the backscatter tag 1 meter off the ground. Accuracy was evaluated using the correct rate.

### 7.2. Experimental Results

The experimental results are as shown in Table 12. In Table 12, it is first observed that in the absence of obstructions, while the accuracy is very low at an overlay number of 10 packets, similar to the evaluation experiments, the accuracy generally exceeds 0.99 when the number of overlays is 50 or more, indicating that the accuracy converges with higher numbers of overlays. Furthermore, in the presence of obstacles, although the accuracy slightly decreases compared to when there are no obstacles, it generally converges to an accuracy above 0.99 when the number of overlays is 50 or more.

Figure 20 and Figure 21 illustrate the detailed changes in accuracy when the number of overlays is varied from 10 to 100. From both figures, it can be observed that the accuracy decreases when the distance between SD-WiFi units is 5 m, regardless of the presence of obstacles. Furthermore, it is evident that the rate at which accuracy converges to above 0.99 is somewhat slower in the presence of obstacles. This can be attributed to the weakening of the backscatter signal due to obstructions. Also, when obstacles are present, the accuracy particularly decreases when the tag is placed at the midpoint of 2.5 m. Similar to the results of Preliminary Experiment 1, this trend can be attributed to the midpoint being more susceptible to the effects of fading.

## 8. Discussion

Initially, Preliminary Experiment 1 evaluated the detection accuracy of the on/off states of frequency-shift backscatter tags in the NAIST Smart Home, which replicated a real living environment. The results indicated that the detection accuracy was generally good for tags located within 1 m of the Tx or Rx, but it decreased for positions more than 1 m away. In particular, the detection accuracy significantly dropped for tags farther from the AP. Hence, in response to RQ1, it can be stated that detection under specific conditions is feasible.

In Preliminary Experiment 2, we evaluated the detection accuracy of frequency-shift backscatter tags attached to everyday objects in the NAIST Smart Home, simulating a real living environment. The results showed that positioning the SD-WiFi AP at a height of 0.75 m or more generally improved accuracy, and a closer vertical distance to the backscatter tag also enhanced detection accuracy. Furthermore, by adjusting the position of the SD-WiFi AP and the backscatter tag, it was possible to achieve an average detection rate of over 90%. Therefore, the answer to RQ2 is that detecting movements of life objects using frequency-shift backscatter tags is feasible under certain conditions.

Evaluation of Experiment 1 demonstrated that typical daily activities could be recognized with high accuracy in experiments with three participants in the NAIST Smart Home. Achieving high accuracy in leave-one-person-out scenarios indicates a low dependency on participants, addressing a common challenge in activity recognition using traditional backscatter technology. Evaluation of Experiment 2 showed that the proposed system and features are effective for in-home activity recognition, even when the activity scenario and duration are variable, achieving high accuracy. Moreover, evaluation of Experiment 3 demonstrated that the proposed method is independent of the room layout, maintaining similar activity recognition accuracy even when the room’s virtual position and activity scenario and duration are variable. Thus, the answer to RQ3 is that in-home activity recognition using frequency-shift backscatter tags is feasible in typical living environments.

On the one hand, relaxing the installation conditions for devices remains a challenge to be addressed in the future. To enhance the detection capabilities of frequency-shift backscatter tags, two potential strategies can be considered. Firstly, increasing the WiFi signal strength is feasible. Currently, in Japan, the maximum transmission power is restricted to 100 mW. If this restriction could be raised, or if applied in other countries like the USA where up to 1 W is possible, it could significantly mitigate the issue of distance. Secondly, increasing the number of packet overlays is necessary. To observe frequency shifts, it is essential to separate them from noise by overlaying multiple packets. Our additional experiments have shown that increasing the number of overlays to approximately 30 to 50 packets can achieve an average detection accuracy of over 0.99, regardless of obstacles. Therefore, enhancing the packet transmission rate or extending the sampling interval could potentially relax the installation constraints of SD-WiFi. The first method provides an advantage when applying our technique in residential settings outside Japan, while the second can be addressed through software improvements.

## 9. Conclusions

In this paper, we implemented frequency-shift backscatter tags-based in-home activity recognition system and investigated its feasibility in a near-real residential setting. Experiments were conducted in the NAIST Smart Home, simulating a real-life living environment.

Preliminary experiments showed that backscatter tags’ states could be detected in real-life environments, albeit with certain limitations regarding the distance between SD-WiFi and the tags. Also, installing backscatter tags on actual daily living objects demonstrated that tag state detection is feasible under specific conditions. The evaluation experiments conducted various activity recognition experiments, demonstrating that adjusting the attenuation rate and window size could achieve high accuracy above 0.95 in all scenarios. In additional experiments, we investigated the feasibility of the extension of the frequency-shifts detection distance by increasing the number of packet overlays. The results revealed that with an overlay number of 30 or more, tags’ states could be detected with an average accuracy exceeding 0.9, regardless of the distance between SD-WiFi units being 3 m or 5 m and the presence of wooden obstructions. This suggests that increasing the number of transmitted packets per second could enable frequency-shift detection in NLoS conditions, potentially allowing for universal activity recognition with a single pair of WiFi APs in domestic settings in the future.

Clarifying the installation conditions for backscatter tags and SD-WiFi in various actual homes is part of our future work. We will focus on relaxing the installation requirements of SD-WiFi through hardware improvements, increasing the packet transmission rate, and adjusting the sampling interval.

## Figures and Tables

**Figure 1 sensors-24-03277-f001:**
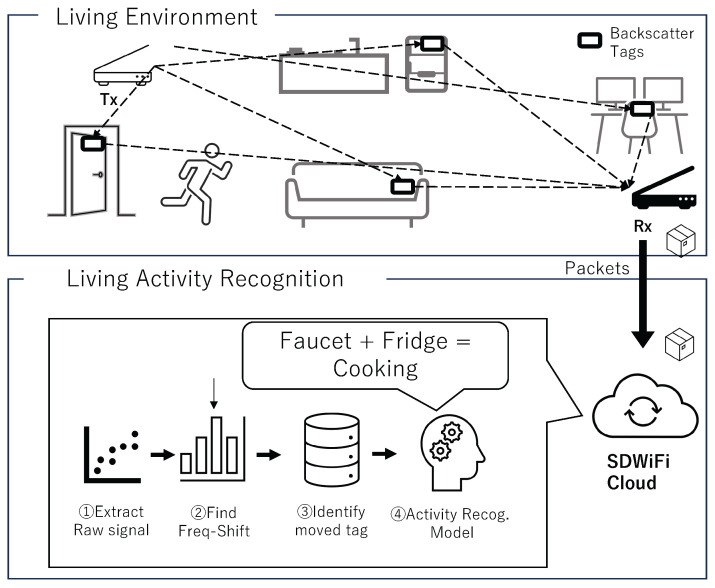
System overview.

**Figure 2 sensors-24-03277-f002:**
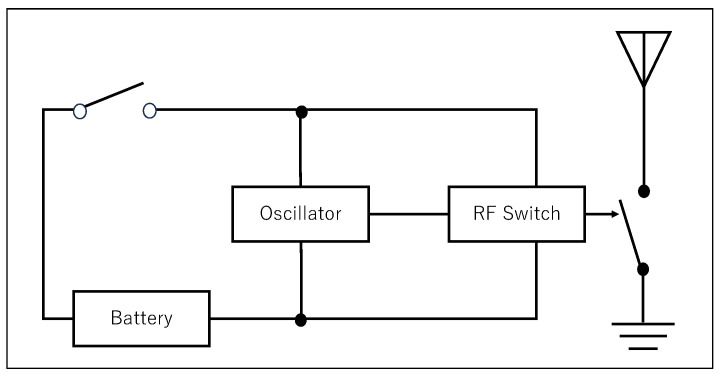
Block diagram of backscatter tag.

**Figure 3 sensors-24-03277-f003:**
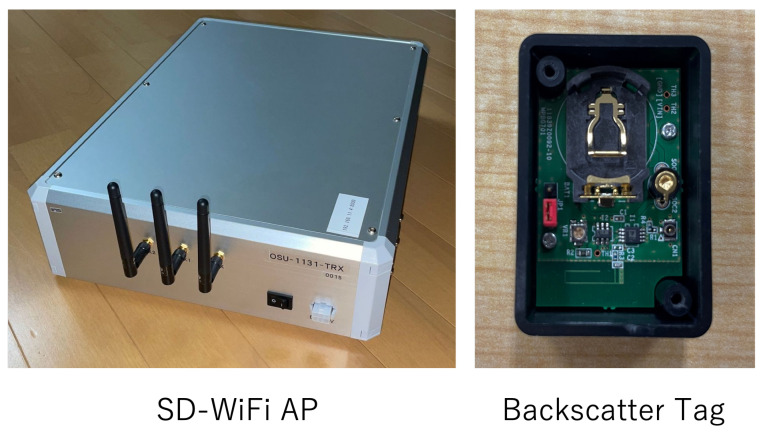
Appearance of SD-WiFi and backscatter tags.

**Figure 4 sensors-24-03277-f004:**
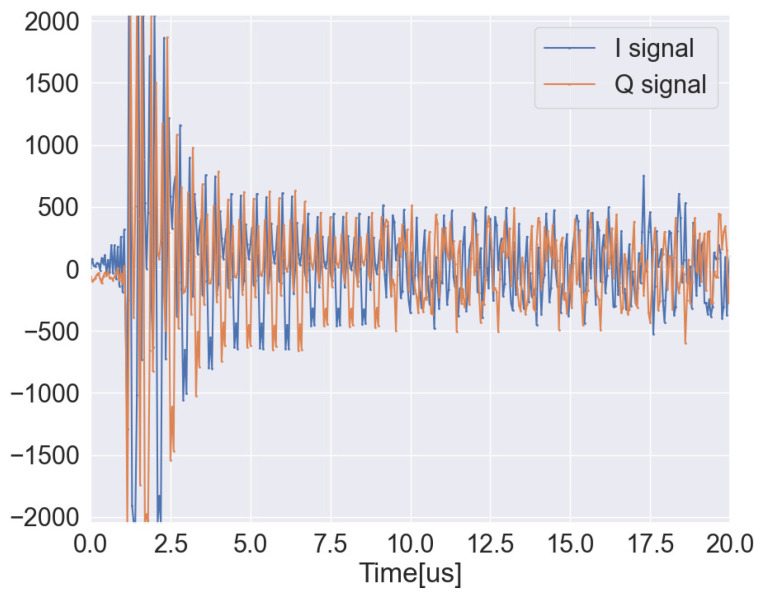
Example of raw signals.

**Figure 5 sensors-24-03277-f005:**
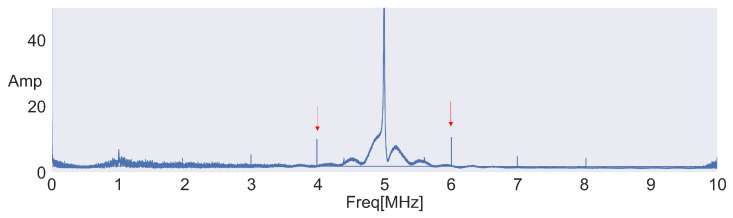
Example of frequency shift.

**Figure 6 sensors-24-03277-f006:**
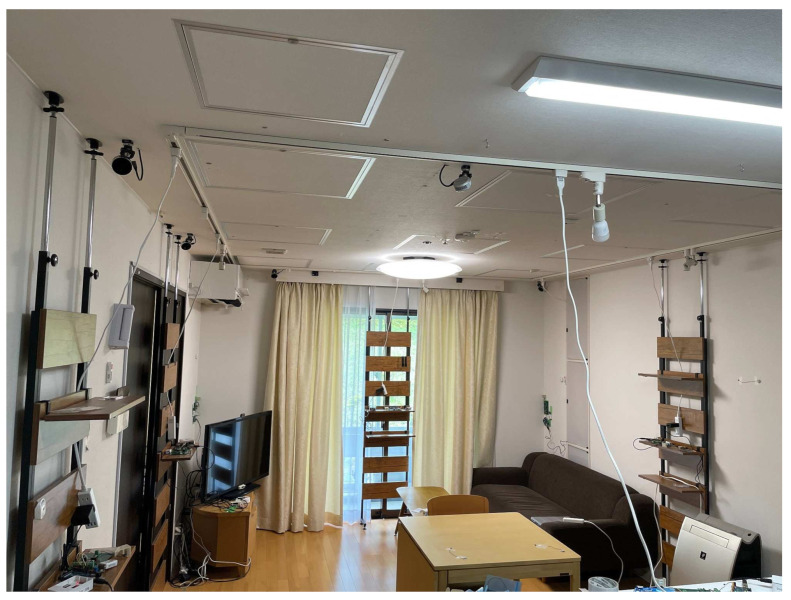
NAIST Smart Home.

**Figure 7 sensors-24-03277-f007:**
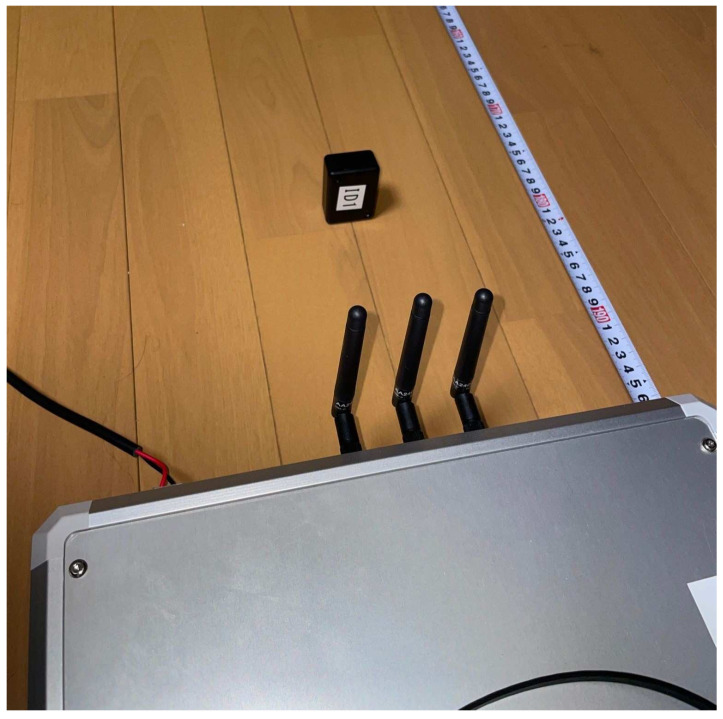
Data collection for Experiment 1.

**Figure 8 sensors-24-03277-f008:**
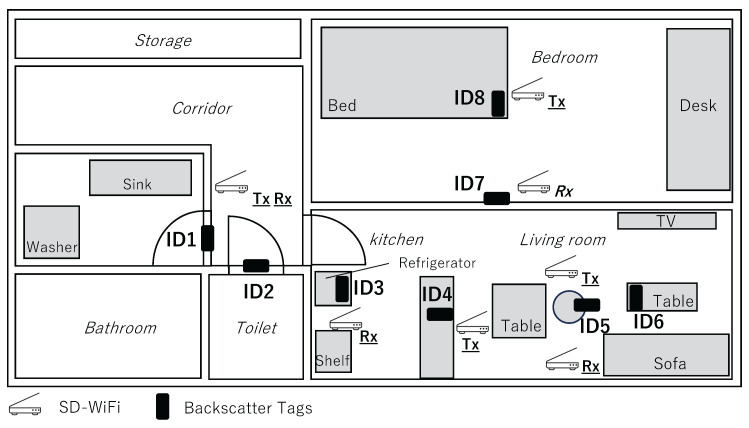
Arrangement of SD-WiFi and backscatter tags.

**Figure 9 sensors-24-03277-f009:**
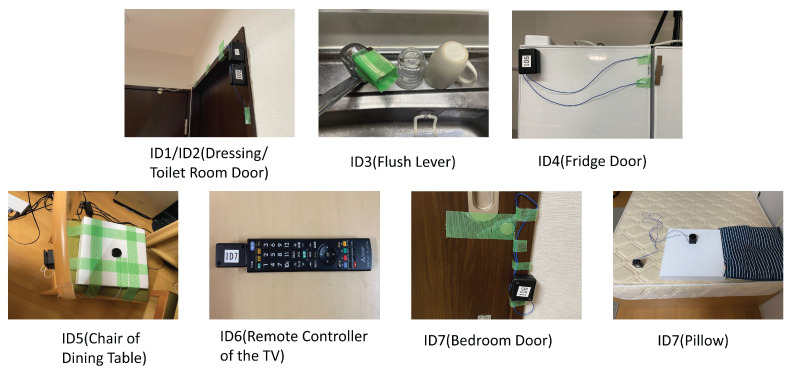
Implementation of backscatter tags.

**Figure 10 sensors-24-03277-f010:**
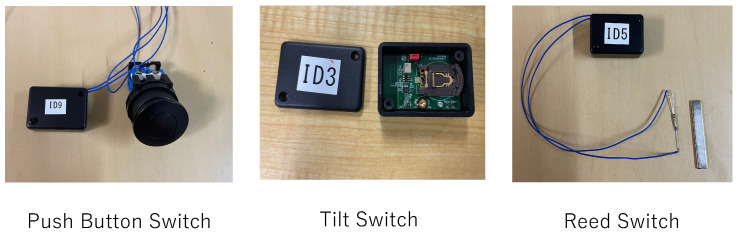
Physical switches.

**Figure 11 sensors-24-03277-f011:**
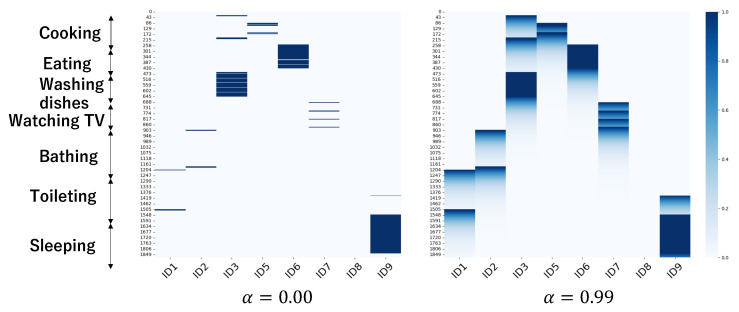
Time series feature matrix (ID1–ID9 in horizontal axis are tags and 0–1850 in vertical axis are seconds elapsed from the beginning of the session; color thickness (0–1.0) means lag features).

**Figure 12 sensors-24-03277-f012:**
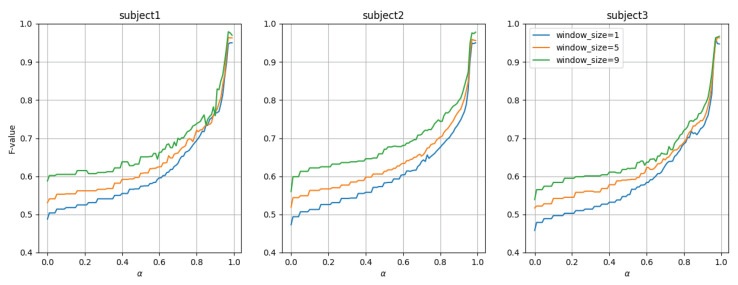
Experiment 1: F-score vs. α (leave-one-session-out cross-validation).

**Figure 13 sensors-24-03277-f013:**
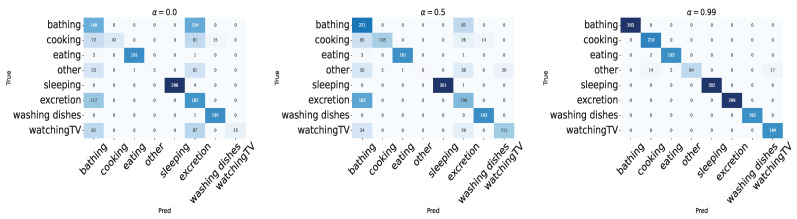
Experiment 1: confusion matrix of leave-one-session-out cross-validation.

**Figure 14 sensors-24-03277-f014:**
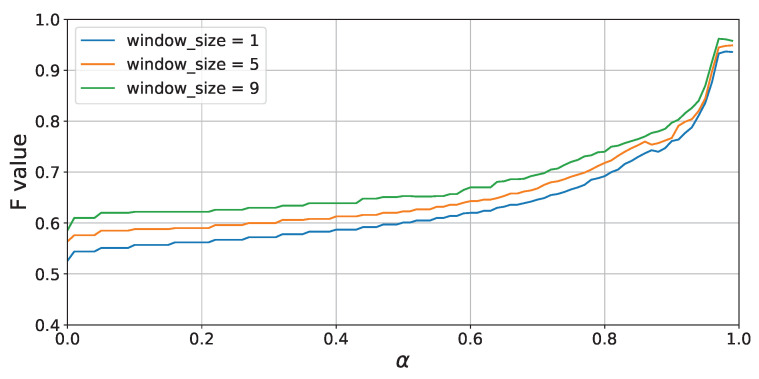
Experiment 1: F-score vs. α (leave-one-person-out cross-validation).

**Figure 15 sensors-24-03277-f015:**
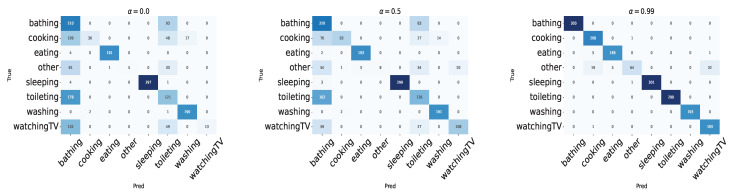
Experiment 1: confusion matrix of leave-one-person-out cross-validation.

**Figure 16 sensors-24-03277-f016:**
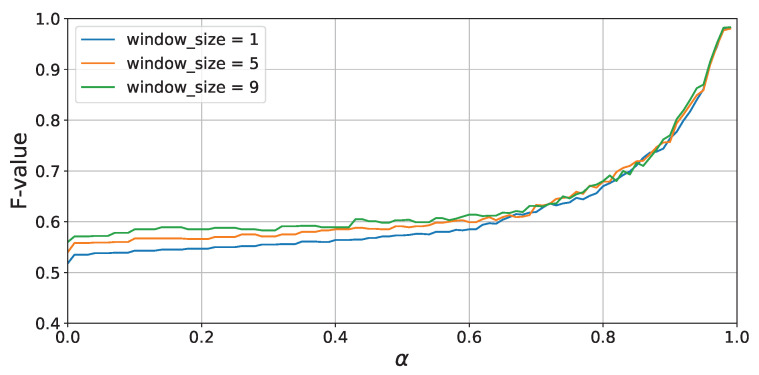
Experiment 2: F-score vs. α (leave-one-session-out cross-validation).

**Figure 17 sensors-24-03277-f017:**
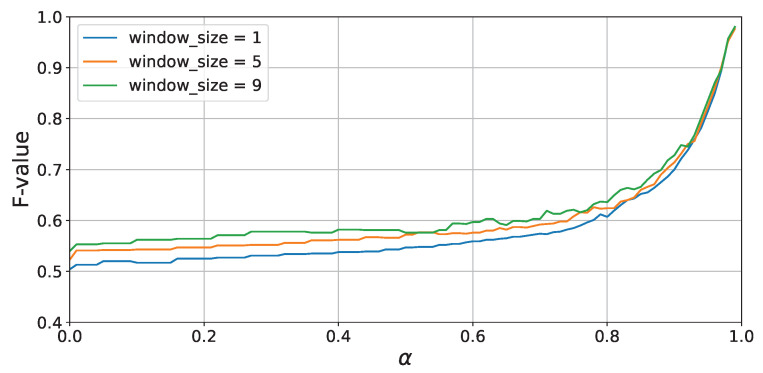
Experiment 3: F-score vs. α (leave-one-session-out cross-validation).

**Figure 18 sensors-24-03277-f018:**
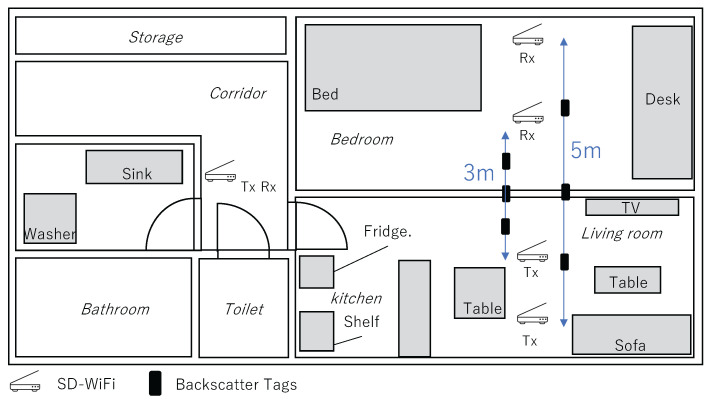
Arrangement of SD-WiFi and backscatter tags.

**Figure 19 sensors-24-03277-f019:**
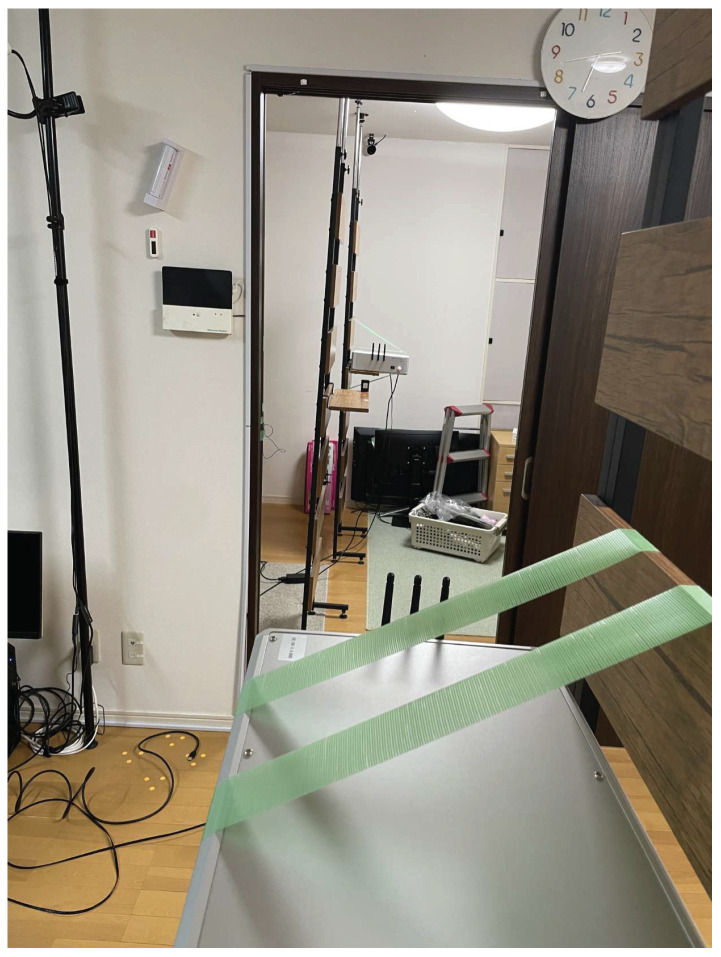
Additional experiments.

**Figure 20 sensors-24-03277-f020:**
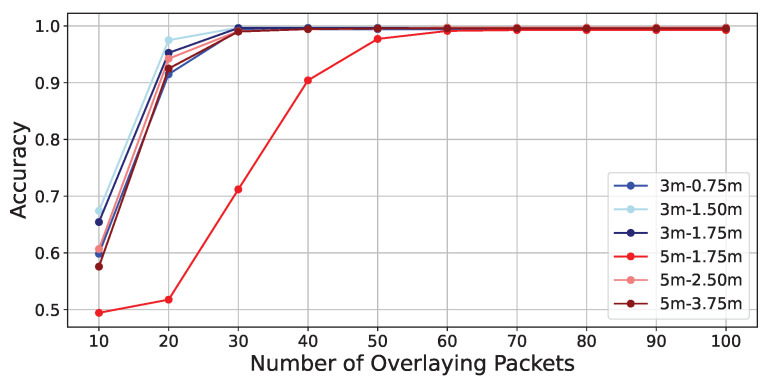
Accuracy vs. packet overlays without-obstructions.

**Figure 21 sensors-24-03277-f021:**
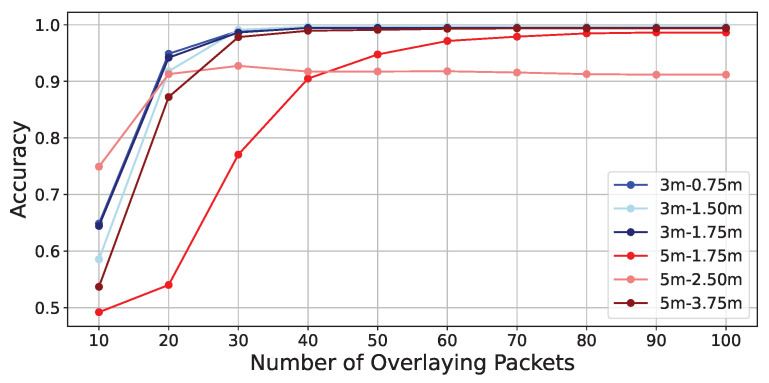
Accuracy vs. packet overlays with-obstructions.

**Table 1 sensors-24-03277-t001:** Detection rate of frequency shift when distance between APs is 1 m.

From Tx	0.25 m	0.50 m	0.75 m
session 1	0.825	0.425	1.000
session 2	1.000	1.000	1.000
session 3	1.000	0.850	1.000
Average	0.942	0.758	1.000

**Table 2 sensors-24-03277-t002:** Detection rate of frequency shift when distance between APs is 2 m.

From Tx	0.25 m	0.50 m	0.75 m	1.00 m	1.25 m	1.50 m	1.75 m
session 1	0.33	1.00	0.33	0.33	0.91	0.33	1.00
session 2	1.00	1.00	1.00	1.00	0.33	0.48	1.00
session 3	1.00	1.00	1.00	0.33	0.39	0.44	1.00
Average	0.78	1.00	0.78	0.62	0.54	0.42	1.00

**Table 3 sensors-24-03277-t003:** Detection rate of frequency shift.

Height	ID1	ID2	ID3	ID4	ID5	ID6	ID7	ID8
0.5 m	-	-	-	-	0.02	0.53	0.05	0.02
0.75 m	-	-	-	-	1.00	1.00	0.01	1.00
1.0 m	0.81	0.00	-	-	0.83	0.71	0.76	0.73
1.25 m	0.12	0.14	0.65	0.80	1.00	0.85	1.00	0.99
1.5 m	0.00	0.33	1.00	1.00	0.00	0.00	0.05	0.98
1.75 m	0.45	0.69	0.86	0.86	0.16	0.66	0.00	1.00
2.0 m	0.98	0.95	0.86	0.93	-	-	0.02	0.92

**Table 4 sensors-24-03277-t004:** Definition of living activities.

Living Activities	Definition
Cooking	Taking food out of the refrigerator then
	raising the faucet lever at the sink to fetch/dump water
Eating	Pulling up a chair and sitting at a desk.
Washing Dishes	Raising the faucet lever at the sink to let the water out.
Watching TV	Turning on the TV and change the channel
Excretion	Opening the toilet door and go inside.
Bathing	Opening the bathroom door and enter
Sleeping	Lying on the bed

**Table 5 sensors-24-03277-t005:** Activity scenario of Experiment 1.

Scenarios	Daily Activity
Scenario 1	Cooking, Eating
Scenario 2	Washing dishes, Watching TV
Scenario 3	Bathing, Toileting, Sleeping

**Table 6 sensors-24-03277-t006:** Scenarios for Experiment 2 and 3.

Scenarios	Session 1	Session 2	Session 3
Scenario 1	Cooking (5 min)	Watching TV (10 min)	Cooking (13 min)
	Eating (10 min)	Cooking (5 min)	Watching TV (5 min)
Scenario 2	Washing Dishes (5 min)	Washing Dishes (3 min)	Eating (8 min)
	Watching TV (10 min)	Eating (5 min)	Washing Dishes (7 min)
Scenario 3	Sleeping (10 min)	Bathing (10 min)	Toileting (3 min)
	Toileting (5 min)	Toileting (10 min)	Bathing (5 min)
	Bathing (8 min)	Sleeping (5 min)	Sleeping (15 min)

**Table 7 sensors-24-03277-t007:** Backscatter tag location and physical switch.

Tags ID	Target	Switch Type
ID1	Dressing Room Door	Reed Switch
ID2	Toilet Door	Reed Switch
ID3	Refrigerator Door	Reed Switch
ID4	Faucet Levers	Tilt Switch
ID5	Chair	Push Button Switch
ID6	Remote Controller of the TV	Tilt Switch
ID7	Bedroom Door	Reed Switch
ID8	Pillow	Push Button Switch

**Table 8 sensors-24-03277-t008:** Experiment 1: leave-one-session-out cross-validation.

Metrics	α=0.0			α=0.5			α=0.99		
	Subject 1	Subject 2	Subject 3	Subject 1	Subject 2	Subject 3	Subject 1	Subject 2	Subject 3
Accuracy	0.630	0.601	0.594	0.681	0.687	0.656	0.972	0.978	0.968
Precision	0.878	0.840	0.828	0.854	0.871	0.812	0.975	0.980	0.972
Recall	0.630	0.601	0.594	0.681	0.687	0.656	0.972	0.978	0.968
F score	0.588	0.560	0.536	0.651	0.671	0.621	0.970	0.978	0.967

**Table 9 sensors-24-03277-t009:** Experiment 1: leave-one-person-out cross-validation.

Metrics	α=0.0	α=0.5	α=0.99
Accuracy	0.586	0.653	0.958
Precision	0.863	0.833	0.964
Recall	0.586	0.653	0.958
F score	0.524	0.621	0.955

**Table 10 sensors-24-03277-t010:** Experiment 2: leave-one-session-out cross-validation.

Metrics	α=0.0	α=0.5	α=0.99
Accuracy	0.632	0.654	0.969
Precision	0.761	0.787	0.975
Recall	0.632	0.654	0.969
F score	0.560	0.592	0.961

**Table 11 sensors-24-03277-t011:** Experiment 3: leave-one-session-out cross-validation.

Metrics	α=0.0	α=0.5	α=0.99
Accuracy	0.624	0.649	0.977
Precision	0.670	0.730	0.972
Recall	0.624	0.649	0.977
F score	0.540	0.576	0.971

**Table 12 sensors-24-03277-t012:** Comparison of experiment results with and without shielding.

Number of Overlaying Packets	Without Shielding						With Shielding					
	3 m			5 m			3 m			5 m		
	0.75 m	1.5 m	2.25 m	1.75 m	2.5 m	3.75 m	0.75 m	1.5 m	2.25 m	1.75 m	2.5 m	3.75 m
10	0.598	0.674	0.654	0.494	0.608	0.576	0.649	0.586	0.644	0.492	0.749	0.530
30	0.992	0.997	0.996	0.712	0.990	0.996	0.989	0.991	0.986	0.770	0.927	0.978
50	0.994	0.997	0.996	0.977	0.995	0.996	0.996	0.967	0.994	0.947	0.917	0.991
80	0.994	0.997	0.996	0.992	0.997	0.996	0.996	0.997	0.994	0.985	0.912	0.994
100	0.994	0.997	0.996	0.992	0.997	0.996	0.996	0.997	0.994	0.986	0.912	0.994
300	0.994	0.997	0.996	0.992	0.997	0.995	0.996	0.997	0.995	0.986	0.894	0.994
500	0.994	0.997	0.996	0.992	0.997	0.995	0.994	0.997	0.996	0.986	0.899	0.995

## Data Availability

Data are contained within the article.

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
