# Peer review of "Daily Living Activity Recognition with Frequency-Shift WiFi Backscatter Tagsâ€"

_sensors, 2024, doi:10.3390/s24113277_

Round 1
Reviewer 1 Report
Comments and Suggestions for Authors
The writing style is clear. Figure 2 is my favorite. My concerns:
- Algorithm 1 is not well-designed.
Foremost, I suggest that you use threshold instead of TH and ∙ (dot multiplication) instead of X in the algorithm. In this way, the algorithm will be more clear at a glance. After all, this is a naive algorithm.
Secondly, using flag instead of result can better reflect this parameter's usage.
Last but not least, there is a massive redundancy in the algorithm presentation:
1. Does it make sense to initially assign average to 0??
2. The loop section of sum is boring and is only suitable for junior programming languages (Basic, Pascal...). What about using sum <- Σarray(0..i+N) that can be understood by all programmers, especially for Python, Matlab, R which are widely used in the field of AI?
3. In fact, even the variable sum could/should be omitted to further intuit the presentation: average <- Σarray(0..i+N) / N.
Man, 6 code lines --> 1 line, but it is even more clear, readable, and saves at least 3 minutes of readers!
"As previously mentioned" should never be used in scientific articles. Please be precise to at least subsection.
- Call for scientific rigor. For example, Table 1 has two flaws:
1. 1.00 and 1.0 exist simultaneously
2. And... What are 1.0 and 1.00??? You can either use 1.000, to mean that you always keep the same number of decimal digits (three) in the same experiment, or use 1, to mean "100% correct" (it's not a joke; it really counts — I could think that 1.000 may also refer to 0.9999, and because there remains uniformly three decimal places, it becomes 1.000 then. But 1 in this case really means "all correct").
The same problem of decimal digit uniform appears in other tables and will not be repeated here.
But. . I still can't help but say that the first column in Table 3 will look much more beautiful if you add some "0"s!
- Although there were only 3 subjects, the authors should still describe their basic conditions thoroughly. For example, you can refer to relevant HAR datasets, such as Germany's CSL-SHARE and the United States' ENABL3S. They all have a very complete presentation of the subject details, as well as the data statistics after collection.
- What is the potential value of this approach to NLOS? Why not conduct (preliminary) experiments in NLOS?
- Pay attention to the traceability of scientific writing. “In this section, we answer the research questions” (raised in Section...)
- The literature review of the manuscript is stale. One example is that [1-10] are all publications that are more than about 5 years old. Are there no latest recognized academic results in these two large fields? It is recommended to refer to and add the latest developments in this field, such as the book "sensors for human activity recognition" (2023) and Yale Hartmann's landmark works on HAR high-level features and other fields from kinematic knowledge, which can be used for any kinds of sensing technologies for HAR (wearables, environment, WiFi...).
The same problem of insufficient/old related work exists in Section 2.1. The results of the past three years have made widespread improvements in the data utilization (removing redundant Wi-Fi antennas), efficiency (using advanced non-depth training recognition to reduce real-time system delay) and accuracy of WiFi ADL recognition, such as last year's "efficient Wi-Fi-Based HAR". Such kinds of up-to-date technologies are fully worthy of the author's reference and application to their algorithm.
Author Response
Dear Reviewer,
Thank you for your comments on the revised draft of our manuscript entitled, "Daily Living Activity Recognition with Frequency-Shift WiFi Backscatter Tags". We have incorporated the changes you have suggested in the revised manuscript and have marked the changed/added texts with red color.
—————————————————————————————————————————
Suggestion 1
Algorithm 1 is not well-designed.
Response 1
We have modified Algorithm1 for easier reading.
The revision is as follows:
- Section 3.3 Activity Recognition Module Algorithm 1
- We summarized the equation for Average in a single line, and used “flag” instead of “result”.
—————————————————————————————————————————
Suggestion 2
"As previously mentioned" should never be used in scientific articles. Please be precise to at least subsection.
Response 2
Thank you for pointing out our writing mistakes. We thoroughly reviewed the article and replaced similar expressions such as “As previously mentioned” and “as mentioned earlier”.
—————————————————————————————————————————
Suggestion 3
Call for scientific rigor. For example, Table 1 has two flaws:
- 1.00 and 1.0 exist simultaneously
- And... What are 1.0 and 1.00??? You can either use 1.000, to mean that you always keep the same number of decimal digits (three) in the same experiment, or use 1, to mean "100% correct" (it's not a joke; it really counts — I could think that 1.000 may also refer to 0.9999, and because there remains uniformly three decimal places, it becomes 1.000 then. But 1 in this case really means "all correct").
The same problem of decimal digit uniform appears in other tables and will not be repeated here.
Response 3
We have corrected the error and standardized the number of digits in each table.
The revision is as follows:
- Table 1 on Page 10
- Table 2 on Page 10
- Table 3 on Page 12
- Table 8 on Page 16
- Table 10 on Page 18
- Table 11 on Page 19
—————————————————————————————————————————
Suggestion 4
Although there were only 3 subjects, the authors should still describe their basic conditions thoroughly. For example, you can refer to relevant HAR datasets, such as Germany's CSL-SHARE and the United States' ENABL3S. They all have a very complete presentation of the subject details, as well as the data statistics after collection.
Response 4
We have added details about the subjects' information to the paper.
The revision is as follows:
- We added detailed information of subjects in section 5.1 Data Collection and Experimental Environment Page 13 Line 393
- “(three males; 24 ± 1 years, 176 ± 2.65cm)”
—————————————————————————————————————————
Suggestion 5
What is the potential value of this approach to NLOS? Why not conduct (preliminary) experiments in NLOS?
Response 5
Firstly, the SDWiFi hardware we currently employ can transmit a maximum of eight packets per second. To achieve real-time activity recognition, considering the window size, it is imperative to detect frequency-shifts using only these eight packets. However, as indicated by the results of additional experiments (Figures 20 and 21), stable detection of frequency shifts in NLoS conditions is challenging with fewer than ten packets. Consequently, preliminary experiments were conducted In LoS and tested layouts that could potentially detect frequency shifts, given the maximum capacity to transmit eight packets per second. The outcomes of additional experiments suggest that increasing the number of transmitted packets per second could enable frequency shift detection in NLoS conditions, potentially allowing for universal activity recognition with a single pair of WiFi APs in domestic settings in the future.
The revision is as follows:
- We added a sentence to clarify the above points in section 9 Conclusion Page 22 Line 633-636
- “This suggests that increasing the number of transmitted packets per second could enable frequency shift detection in NLoS conditions, potentially allowing for universal activity recognition with a single pair of WiFi APs in domestic settings in the future.”
—————————————————————————————————————————
Suggestion 6
Pay attention to the traceability of scientific writing. “In this section, we answer the research questions” (raised in Section...)
Response 6
We removed the sentence “In this section, we answer the research questions.” in section 8.
—————————————————————————————————————————
Suggestion 7
The literature review of the manuscript is stale. One example is that [1-10] are all publications that are more than about 5 years old. Are there no latest recognized academic results in these two large fields? It is recommended to refer to and add the latest developments in this field, such as the book "sensors for human activity recognition" (2023) and Yale Hartmann's landmark works on HAR high-level features and other fields from kinematic knowledge, which can be used for any kinds of sensing technologies for HAR (wearables, environment, WiFi...).
Response 7
Thank you for your comments. Our objective is to conduct daily living activity recognition within households using novel sensor technologies. The papers cited as [1-10] are essential for illustrating the evolution and challenges in traditional research on general-purpose activity recognition within homes. However, recognizing that many of these references are somewhat dated, we have updated some references.
The revision is as follows:
- Section 1 Introduction Page 1 Line 24-25
- “Methods for recognizing daily activities can be broadly categorized into two approaches: those that require wearing devices such as smartphones and wearable devices [1–9], and those that involve installing non-contact sensors in the environment [10–13].”
- Section 2.2 WiFi-based Methods Page 1 Line 24-25
- “Jannat et al. present a Wi-Fi-based human activity recognition system that uses an Adaptive Antenna Elimination algorithm, achieving accuracies up to 99.84% on activities like walking, falling, and sitting across different environments [21].Ding et al. introduced a device-free human activity recognition system utilizing a deep complex network that processes both amplitude and phase of Wi-Fi signals. Tested in an office
environment, this method demonstrated high accuracies of 96.85% and 94.02% across 24 locations for five distinct activities [22].”
—————————————————————————————————————————
Again, thank you for giving us the opportunity to strengthen our manuscript with your valuable
comments and queries. We have worked hard to incorporate your feedback and hope that
these revisions will persuade you to accept our submission.
Sincerely,
Hikoto Iseda
Graduate School of Information Science, Nara Institute of Science and Technology
8916-5, Takayama, Ikoma, Nara, 630-0192, JAPAN
iseda.hikoto.ih4@is.naist.jp

Reviewer 2 Report
Comments and Suggestions for Authors
This paper implemented a frequency-shift backscatter tags-based in-home activity recognition system and investigated its feasibility in a near-real residential setting. The system consists of SD-WiFi, a software-defined WiFi AP, and physical switches on backscatter tags tailored for detecting the movements of daily objects. Experiments were conducted in the NAIST Smart Home with 96% accuracy in recognizing seven typical daily living activities with an appropriate transceiver layout. Overall, the paper leans more toward experimentation rather than technical exposition.
Strengths:
+ The idea of using the frequency-shift backscatter tag to achieve in-home activity recognition is very interesting.
+ The authors have conducted comprehensive experiments to investigate the performance of the system in a near-real environment.
+ This paper is clear and well-structured.
Weakness:
+ The paper is not well motivated. The method proposed does not fully address the pain points of other techniques mentioned in the motivation section, such as RFID technology requiring specialized equipment.
+ The article fails to delve into critical technical details, such as why the tag's ON/OFF modulation leads to frequency shifts and the specific feature patterns for different activities.
+ The article mainly focuses on experimentation, resulting in a lack of innovation, with most of its content merely focusing on introducing and experimentally verifying existing technologies in various environmental conditions.
Detailed comments:
+ The motivation of this paper is not clear. The limitations of WiFi CSI and RFID technologies, such as restricted distance, are highlighted in the motivation section of the article. However, the methods proposed in the article also have a similarly limited applicable range, not exceeding 5 meters, failing to effectively address this constraint.
+ The article points out another drawback of RFID technology: the requirement for specialized equipment, which thereby limits its practical applications in real-world scenarios. However, the devices proposed in the article also require specialized equipment rather than commercial Wi-Fi devices, specifically software-defined radio-based Wi-Fi APs.
+ In the section discussing activity recognition, there is no explanation provided for why the tags only respond during specific actions and remain unresponsive at other times. Moreover, different activities should have distinct characteristic patterns. It should be clearly explained how random forest learning is utilized, whether templates need to be measured first, and other related details. It is suggested to include specific figures depicting the features collected from various activities rather than directly presenting the test results.
+ The equations in the article are not numbered, and the images lack sub-figure numbering. For example, in the first equation on page 6, when representing a signal using sine and cosine functions, the frequency should be multiplied by 2 pi.
Comments on the Quality of English Language
This paper is well-written and easy to understand.
Author Response
Dear Reviewer,
Thank you for your comments on the revised draft of our manuscript entitled, "Daily Living Activity Recognition with Frequency-Shift WiFi Backscatter Tags". We have incorporated the changes you have suggested in the revised manuscript and have marked the changed/added texts with red color.
—————————————————————————————————————————
Suggestion 1
The motivation of this paper is not clear. The limitations of WiFi CSI and RFID technologies, such as restricted distance, are highlighted in the motivation section of the article. However, the methods proposed in the article also have a similarly limited applicable range, not exceeding 5 meters, failing to effectively address this constraint.
Response 1
As you pointed out, the experiments conducted in this paper did not fully resolve the limitations regarding distance. This issue stems from the current software constraints of our SD-WiFi AP, which limits the number of packets transmitted to a maximum of eight per second. However, our additional experiments demonstrated that by increasing the number of packet overlays, sensing is feasible at greater distances and in non-line-of-sight (NLoS) conditions. Therefore, by developing software that increases packet transmission speeds and adjusting the frequency shift detection intervals, we anticipate that future implementations will enable comprehensive in-home sensing using only a pair of devices.
The revision is as follows:
- We added a sentence to clarify the above points in section 9 Conclusion Page 22 Line 633-636
- “This suggests that increasing the number of transmitted packets per second could enable frequency shift detection in NLoS conditions, potentially allowing for universal activity recognition with a single pair of WiFi APs in domestic settings in the future.”
—————————————————————————————————————————
Suggestion 2
The article points out another drawback of RFID technology: the requirement for specialized equipment, which thereby limits its practical applications in real-world scenarios. However, the devices proposed in the article also require specialized equipment rather than commercial Wi-Fi devices, specifically software-defined radio-based Wi-Fi APs.
Response 2
In our research, we utilized SD-WiFi, which transmits sensing-specific Continuous Wave (CW) signals within the payload section of WiFi packets while adhering to the standard WiFi packet format. Generating CW signals is a simpler function compared to the generation of WiFi's OFDM signals; therefore, we believe that the implementation within WiFi systems is straightforward. Currently, specialized equipment is required to generate CW signals. However, this research demonstrates the effectiveness of CW-based sensing, leading us to anticipate its potential adoption in future WiFi sensing standards, such as those proposed in the ongoing IEEE 802.11bf standardization process.
—————————————————————————————————————————
Suggestion 3
In the section discussing activity recognition, there is no explanation provided for why the tags only respond during specific actions and remain unresponsive at other times. Moreover, different activities should have distinct characteristic patterns. It should be clearly explained how random forest learning is utilized, whether templates need to be measured first, and other related details. It is suggested to include specific figures depicting the features collected from various activities rather than directly presenting the test results.
Response 3
The backscatter tags we used are connected to simple physical switches, which activate to "ON" when human interactions occur with furniture, appliances, or doors, thus completing the circuit and generating a backscatter signal that results in a frequency shift, as depicted in Figure 2. The activity recognition module utilizes a bandpass filter to only observe whether peaks exist within a specific frequency band. This process underlies why the on/off states of the tags are linked to specific activities. For example, the activity "toileting" is associated with the tag on the toilet door. By using machine learning to automatically learn these associations, we enable activity recognition.
To clarify these points further, we have added a diagram visualizing the features and a description of this visualization to the paper.
The revision is as follows:
- We have added a figure to illustrate feature extraction in Page 15
- Add following statement in Page 15 Line 448-462
- “In Figure 11, we present the scenarios without the introduction of lag features (a=0.00) and with the introduction of lag features (a=0.99). This figure illustrates that the introduction of lag features results in more complex input features.}”
—————————————————————————————————————————
Suggestion 4
The equations in the article are not numbered, and the images lack sub-figure numbering. For example, in the first equation on page 6, when representing a signal using sine and cosine functions, the frequency should be multiplied by 2 pi.
Response 4
Thank you for your comments. I have now numbered all equations and corrected the error in the equation that explains the frequency shift.
The revision is as follows:
- we added equation numbers
- Page 6 Line 222
- Page 7 Line 259
- Page 8 Line 272
- Page 15 Line 458
- We fixed Equation 1, which explains the frequency shift in Page 6 Line 222
- 2sin(2πft)sin(2πf_it) = cos(2π(f−f_i)t) − cos(2π( f + f_i)t)
—————————————————————————————————————————
Again, thank you for giving us the opportunity to strengthen our manuscript with your valuable
comments and queries. We have worked hard to incorporate your feedback and hope that
these revisions will persuade you to accept our submission.
Sincerely,
Hikoto Iseda
Graduate School of Information Science, Nara Institute of Science and Technology
8916-5, Takayama, Ikoma, Nara, 630-0192, JAPAN
iseda.hikoto.ih4@is.naist.jp

Round 2
Reviewer 1 Report
Comments and Suggestions for Authors
The revision is satisfactory. There are still two points that need improvement:
1. Still Algorithm 1. There should be a break after flag<-- True. This can reduce the time complexity of the algorithm from O(N) to at least N/2, or even logN or sqrtN, depending on the data peak distribution. See, although this is just a very short and compact algorithm, there are so many locations where it can be optimized! Maybe you have used break in your actual programming but not written in the manuscript. If not in practice, you could (should) add it. A saving of 0.001 milliseconds at a time may add up to minutes/hours of time savings for computing large amounts of data, not to mention in the future you will expand the data volume and even use multiple datasets.
2. I firmly believe that Hartmann’s high-level HAR feature design (https://link.springer.com/chapter/10.1007/978-3-031-38854-5_8) has considerable reference significance for your work, although it is not a direct template for your work. It should be discussed.
Author Response
Response letter (to Reviewer 1)
Dear Reviewer,
Thank you for your comments on the revised draft of our manuscript entitled, "Daily Living Activity Recognition with Frequency-Shift WiFi Backscatter Tags". We have incorporated the changes you have suggested in the revised manuscript and have marked the changed/added texts with red color.
—————————————————————————————————————————
Suggestion 1
Still Algorithm 1. There should be a break after flag<-- True. This can reduce the time complexity of the algorithm from O(N) to at least N/2, or even logN or sqrtN, depending on the data peak distribution. See, although this is just a very short and compact algorithm, there are so many locations where it can be optimized! Maybe you have used break in your actual programming but not written in the manuscript. If not in practice, you could (should) add it. A saving of 0.001 milliseconds at a time may add up to minutes/hours of time savings for computing large amounts of data, not to mention in the future you will expand the data volume and even use multiple datasets.
Response 1
Thank you for your comment. Indeed, it is more rational to introduce a break at the point where the flag becomes true. We will revise Algorithm 1 accordingly.
The revision is as follows:
- Section 3.3 Activity Recognition Module Algorithm 1 in Page 8 Line 282
- We inserted “break” at line 7 of algorithm 1.
—————————————————————————————————————————
Suggestion 2
I firmly believe that Hartmann’s high-level HAR feature design (https://link.springer.com/chapter/10.1007/978-3-031-38854-5_8) has considerable reference significance for your work, although it is not a direct template for your work. It should be discussed.
Response 2
Thank you for suggesting the valuable paper. Indeed, it seems to be useful for our research, hence it has been added to our references.
The revision is as follows:
- Section 1 Introduction Page 1 Line 26-29
- This method is generally capable of tracking fine-grained bodily movements with high accuracy, and effective feature extraction techniques have been established, as exemplified by Hartmann et al. However, it poses a significant physical burden on users [14].
—————————————————————————————————————————
Again, thank you for giving us the opportunity to strengthen our manuscript with your valuable comments and queries. We have worked hard to incorporate your feedback and hope that these revisions will persuade you to accept our submission.
Sincerely,
Hikoto Iseda
Graduate School of Information Science, Nara Institute of Science and Technology
8916-5, Takayama, Ikoma, Nara, 630-0192, JAPAN
iseda.hikoto.ih4@is.naist.jp

Reviewer 2 Report
Comments and Suggestions for Authors
The authors have addressed most of my concerns. However, the motivation and the difference from the existing solutions are not clear to me.
- For the motivation, the authors claim that by developing software that increases packet transmission speeds and adjusting the frequency shift detection intervals, they anticipate that future implementations can improve the distance. I think the future implementations of WiFi CSI and RFID may also get longer sensing distance with better hardware and software. Therefore, such a motivation is not stronger for the reader. It is important to add a comprehensive discussion to show how to improve the distance.
- The authors agree that this work also require the specialized equipment to generate CW signal. Therefore, the specialized equipment is not the advantage compared with the existing RFID technology, which also require the specialized equipment. If so, the authors need to delete these part.
Comments on the Quality of English LanguageThis paper is well-written and easy to understand.
Author Response
Response letter (to Reviewer 2)
Dear Reviewer,
Thank you for your comments on the revised draft of our manuscript entitled, "Daily Living Activity Recognition with Frequency-Shift WiFi Backscatter Tags". We have incorporated the changes you have suggested in the revised manuscript and have marked the changed/added texts with red color.
—————————————————————————————————————————
Suggestion 1
For the motivation, the authors claim that by developing software that increases packet transmission speeds and adjusting the frequency shift detection intervals, they anticipate that future implementations can improve the distance. I think the future implementations of WiFi CSI and RFID may also get longer sensing distance with better hardware and software. Therefore, such a motivation is not stronger for the reader. It is important to add a comprehensive discussion to show how to improve the distance.
Response 1
While WiFi Channel State Information (CSI) can potentially extend detection ranges, it fundamentally suffers from environmental dependency. In contrast, RFID-based methods require specialized RFID readers, which are not widely adopted in general households. Our approach solely detects frequency shifts and does not target the radio environment itself, thus minimizing environmental dependency. Moreover, WiFi Access Points (APs) are already prevalent in many homes, providing a readily available infrastructure.
To enhance the detection capabilities of frequency-shift backscatter tags, two strategies can be applied. First, increasing the WiFi signal strength could be beneficial. Currently, the maximum transmission power in Japan is limited to 100mW. If this limit were raised, or in other countries like the USA where up to 1W is possible, the issue of distance could be significantly alleviated. Second, increasing the number of packet overlays is crucial. Observing frequency shifts requires separating them from noise by overlaying multiple packets; our additional experiments demonstrated that more overlays allow for detecting even weak frequency shifts. Therefore, increasing the packet transmission rate or extending the sampling interval could relax the installation constraints of SD-WiFi. The first method offers an advantage when applying our technique in residential settings outside Japan, while the second can be addressed through software improvements.
To clarify these points, the following modifications have been made to the paper:
- We added a sentence to clarify the above points in section 8 Discussion Page 23 Line 603-616
- “On the one hand, relaxing the installation conditions for devices remains a challenge to be addressed in the future. To enhance the detection capabilities of frequency-shift backscatter tags, two potential strategies can be considered. Firstly, increasing the WiFi signal strength is feasible. Currently, in Japan, the maximum transmission power is restricted to 100mW. If this restriction could be raised, or if applied in other countries like the USA where up to 1W is possible, it could significantly mitigate the issue of distance. Secondly, increasing the number of packet overlays is necessary. To observe frequency shifts, it is essential to separate them from noise by overlaying multiple packets. Our additional experiments have shown that increasing the number of overlays to approximately 30 to 50 packets can achieve an average detection accuracy of over 0.99, regardless of obstacles. Therefore, enhancing the packet transmission rate or extending the sampling interval could potentially relax the installation constraints of SD-WiFi. The first method provides an advantage when applying our technique in residential settings outside Japan, while the second can be addressed through software improvements.”
—————————————————————————————————————————
Suggestion 2
The authors agree that this work also require the specialized equipment to generate CW signal. Therefore, the specialized equipment is not the advantage compared with the existing RFID technology, which also require the specialized equipment. If so, the authors need to delete these part.
Response 2
RFID readers are not yet widely adopted in general households and are unlikely to become as ubiquitous as WiFi in the foreseeable future. This is particularly true in activity recognition applications that require the capability to scan multiple RFID tags simultaneously, as devices that perform this function have very limited applications, thus offering little incentive for widespread home adoption. In contrast, WiFi access points (APs) serve a practical purpose by connecting user devices like smartphones to the internet, and are already prevalent in many homes. In this sense, WiFi APs are not as specialized as RFID readers. Indeed, the SD-WiFi we utilize incorporates specialized devices for generating Continuous Wave (CW) signals, and although this requires an extension beyond standard WiFi AP functionalities, it still complies with WiFi protocols. Our research is focused on establishing a versatile method for activity recognition in typical households. Therefore, compared to RFID, the barriers to widespread adoption of WiFi APs equipped with CW signal capabilities are considerably lower in residential settings.
To clarify these points, the following modifications have been made to the paper:
- We added a sentence to clarify the above points in section 1 Introduction Page 2 Line 70-78
- We utilize SD-WiFi, a software-defined WiFi access point, as the communication foundation, which can transmit packets containing Continuous Wave (CW) signals as part of the payload. While such specifications are indeed not found in conventional WiFi devices, unlike RFID readers which are not commonly deployed in most households, WiFi devices for transmitting and receiving are already widespread in general households. This makes them well-suited as a foundation for recognizing daily activities in residential environments. Also, unlike traditional methods, SD-WiFi enables discrete communication on a per-packet basis, facilitating coexistence with various other signals in real residential environments.
—————————————————————————————————————————
Again, thank you for giving us the opportunity to strengthen our manuscript with your valuable
comments and queries. We have worked hard to incorporate your feedback and hope that
these revisions will persuade you to accept our submission.
Sincerely,
Hikoto Iseda
Graduate School of Information Science, Nara Institute of Science and Technology
8916-5, Takayama, Ikoma, Nara, 630-0192, JAPAN
iseda.hikoto.ih4@is.naist.jp
